Plasma BDNF concentrations and the antidepressant effects of six ketamine infusions in unipolar and bipolar depression

Zheng Wei zhengwei0702@163.com 1
Zhou Yan-Ling 1
Wang Cheng-Yu 1
Lan Xiao-Feng 1
Zhang Bin 1
Zhou Su-Miao 1
Yan Su 1
Ning Yu-Ping ningjeny@126.com 1 2
1 Psychiatry, The Affiliated Brain Hospital of Guangzhou Medical University (Guangzhou Huiai Hospital) , Guangzhou , China
2 Psychiatry, The first School of Clinical Medicine, Southern Medical University , Guangzhou , China
Zhong Bao-Liang
Electronic publication date: 2021 Mar 29
Publication date: 2021
Volume: 9
Electronic Location ID: e10989
Received 2020 Nov 24; Accepted 2021 Feb 1
Copyright: ©2021 Zheng et al.
Copyright year: 2021
Copyright holder: Zheng et al.
License: This is an open access article distributed under the terms of the Creative Commons Attribution License, which permits unrestricted use, distribution, reproduction and adaptation in any medium and for any purpose provided that it is properly attributed. For attribution, the original author(s), title, publication source (PeerJ) and either DOI or URL of the article must be cited.
License URL: https://creativecommons.org/licenses/by/4.0/

Keywords: Ketamine, Brain-derived neurotrophic factor, Depression, Predictors, Response

Funding: Science and Technology Planning Project of Liwan District of Guangzhou 202004034 Guangzhou Health Science and Technology Project 20211A011045 Guangzhou science and Technology Project of traditional Chinese Medicine and integrated traditional Chinese and Western medicine 20211A011045 China International Medical Exchange Foundation Z-2018-35-2002 Guangdong Hospital Association 2019ZD06 This study was funded by the Science and Technology Planning Project of Liwan District of Guangzhou (202004034), Guangzhou Health Science and Technology Project (20211A011045), Guangzhou science and Technology Project of traditional Chinese Medicine and integrated traditional Chinese and Western medicine (20211A011045), China International Medical Exchange Foundation (Z-2018-35-2002), Guangzhou Clinical Characteristic Technology Project (2019TS67), and Guangdong Hospital Association (2019ZD06). The funders had no role in study design, data collection and analysis, decision to publish, or preparation of the manuscript.

==============================
Objectives

Accumulating evidence has implicated that brain derived neurotrophic factor (BDNF) is thought to be involved in the pathophysiology of depression, but its correlation with ketamine’s antidepressant efficacy focusing on Chinese individuals with depression is not known. This study was aim to determine the correlation of plasma BDNF (pBDNF) concentrations and ketamine’s antidepressant efficacy.

Methods

Ninety-four individuals with depression received six intravenous infusions ketamine (0.5 mg/kg). Remission and response were defined as Montgomery-Asberg Depression Rating Scale (MADRS) scores less than 10 and a reduction of 50% or more in MADRS scores, respectively. Plasma was collected at baseline and at 24 h and 2 weeks after completing six ketamine infusions (baseline, 13 d and 26 d).

Results

A significant improvement in MADRS scores and pBDNF concentrations was found after completing six ketamine infusions compared to baseline (all ps < 0.05). Higher baseline pBDNF concentrations were found in ketamine responders/remitters (11.0 ± 6.2/10.1 ± 5.8 ng/ml) than nonresponders/nonremitters (8.0 ± 5.5/9.2 ± 6.4 ng/ml) (all ps < 0.05). Baseline pBDNF concentrations were correlated with MADRS scores at 13 d (t =  − 2.011, p = 0.047) or 26 d (t =  − 2.398, p = 0.019) in depressed patients (all ps < 0.05). Subgroup analyses found similar results in individuals suffering from treatment refractory depression.

Conclusion

This preliminary study suggests that baseline pBDNF concentrations appeared to be correlated with ketamine’s antidepressant efficacy in Chinese patients with depression.

Introduction

Accumulating evidence suggests that glutamatergic abnormalities are associated with the pathophysiology of mood disorders (Yüksel & Öngür, 2010). Numerous early studies had consistently reported that an antagonist of glutamatergic N-methyl-D-aspartate (NMDA) receptors ketamine at subanesthetic doses could result in fast-acting and sustained antidepressant effects in individuals suffering from unipolar and bipolar depression (Na & Kim, 2021; Phillips et al., 2020). For example, ketamine’s repeated administration had quick and enduring antidepressant and antisuicidal effects in depressed patients (Kryst et al., 2020).

The precise mechanisms underlying subanesthetic intravenous ketamine’s antidepressant actions are still incompletely understood (Rong et al., 2018). A recent animal study found that blockade of NMDA receptors increased the induction of α-amino-3-hydroxy-5-methyl-4-isoxazole propionic acid (AMPA) receptor expression in models of depression, and subsequent activation of the mammalian target of rapamycin (mTOR) pathway was needed for the rapid and robust antidepressant action of ketamine (Li et al., 2010). Growing evidence implicated neurotrophic factors, such as brain-derived neurotrophic factor (BDNF), played an important in the pathophysiology of mood disorders (Duman, 2004; Duman & Monteggia, 2006). BDNF is a key protein in facilitating and supporting memory growth and neuronal survival (Leal, Bramham & Duarte, 2017). Rapid and transient upregulation of BDNF reversed or blocked atrophy and cell loss in patients with depression, and it may be a critical component in subanesthetic intravenous ketamine’s antidepressant actions (Haile et al., 2014).

In general, BDNF plays a role in the pathophysiology of schizophrenia (Singh et al., 2020) and mood disorders (Molendijk et al., 2014; Sagud et al., 2016). For example, several studies found that individuals suffering from depression had lower serum BDNF concentrations and pBDNF concentrations than that of healthy subjects (Molendijk et al., 2014; Sagud et al., 2016) and recovered after successful antidepressant therapy (Brunoni, Lopes & Fregni, 2008; Polyakova et al., 2015). Central and peripheral BDNF is positively correlated with the response and remission of antidepressant treatment (Lee & Kim, 2010). Notably, Kurita et al. (2012) reported that remitted than nonremitted depressed patients appeared to have higher pBDNF concentrations, and these concentrations were associated with the Montgomery-Asberg Depression Rating Scale (MADRS) scores.

BDNF as a predictor of ketamine’s antidepressant efficacy in individuals suffering from treatment-refractory depression (TRD) has been investigated, but with inconsistent findings. For example, several open-label studies on ketamine and BDNF found a negative association of the increase in BDNF following a single ketamine infusion with the severity of depression (Cornwell et al., 2012; Duncan et al., 2013). Another study found that BDNF did not mediate single subanesthetic intravenous ketamine’s antidepressant efficacy (Machado-Vieira et al., 2009). However, no studies had been published to examine the relationship of pBDNF concentrations and serial subanesthetic intravenous ketamine infusions’ antidepressant efficacy in Chinese individuals suffering from depression.

The present study was performed to examine the correlation of pBDNF concentrations and six subanesthetic intravenous ketamine’s antidepressant efficacy (0.5 mg/kg) administered thrice weekly over two weeks in Chinese individuals suffering from unipolar and bipolar depression. In this study, we hypothesized that serial intravenous subanesthetic ketamine would increase pBDNF concentrations, and baseline pBDNF concentrations would be associated with ketamine’s antidepressant efficacy in individuals suffering from depression.

Methods

Study sample

Data of the current study were collected from an open-label clinical study, which examined serial intravenous subanesthetic ketamine’s antidepressant and antisuicidal efficacy in individuals suffering from depression and was performed between November 2016 to December 2017 (registration number: ChicCTR-OOC-17012239) (Zheng et al., 2018). The Ethics Committee of the Affiliated Brain Hospital of Guangzhou Medical University approved the current trial’s protocol (Ethical Application Ref: 2016-030) and written informed consent was obtained from all participants.

All subjects were recruited based on the following inclusion criteria: (1) aged between 18 and 65 years, without psychotic symptoms; (2) diagnosis of unipolar or bipolar depression by a certified psychiatrist according to the Structured Clinical Interview for DSM-V (SCID-5) criteria, with a score of 17 or more for the Hamilton Depression Rating Scale (HAMD-17) (Hamilton, 1960); (3) suffering from TRD, which was defined as nonresponse to 2 or more antidepressant treatments, or experiencing suicidal ideation as measured with the Scale for Suicidal Ideations (Beck, Kovacs & Weissman, 1979); (4) had no a history of neurological diseases (e.g., dementia), drug or alcohol abuse; (5) negative urine toxicology; (6) were not pregnant or breast feeding; and (7) had no any unstable medical illness (e.g., cerebrovascular diseases).

Treatment

All patients received a thrice-weekly ketamine treatment regimen for 2 weeks, with a follow-up period of two weeks. The method for repeated ketamine infusions was described in detail in our early trial (Zheng et al., 2018). Briefly, vital signs and clinical status of participants were routinely monitored, and each subject received six intravenous infusions of 0.5 mg/kg ketamine over 40 min. During the study period, all subjects continued taking psychotropic agents.

Response and remission

The MADRS (Montgomery & Asberg, 1979; Zhong et al., 2011) was used to assess depressive symptoms at baseline, 1 d after the sixth infusion (13 d), and 2 weeks after the last ketamine treatments (26 d). Remission and response were defined as MADRS scores less than 10 (Zimmerman, Posternak & Chelminski, 2004) and a reduction of 50% or more in MADRS scores, respectively.

Measurement of pBDNF concentrations

Plasma was collected at baseline, 13 d and 26 d, which were stored at −80 °C until further use. In accordance with the manufacturer’s instructions, in this study a commercially available enzyme-linked immunosorbent assay (ELISA) kit (EMD Millipore Corporation, MA, USA) was used to measure pBDNF concentrations.

Statistical analysis

The Mann–Whitney U test was conducted to analyze nonnormally distributed continuous data, and independent t tests were applied for normally distributed continuous data. For categorical variables, the Fisher’s exact test or Chi-squared test were applied for comparisons between groups (responders versus nonresponders and remitters versus nonremitters). Changes in pBDNF concentrations and MADRS scores over time and subgroup differences (responders/nonresponders and remitters/nonremitters) were examined using linear mixed models. Bivariate correlation analysis was applied in order to determine the correlation of baseline pBDNF concentrations and MADRS scores at 13 d and 26 d in individuals suffering from unipolar or bipolar depression. Multiple linear regression were also used to examine the independent association of baseline pBDNF concentrations and MADRS scores at 13 d and 26 d. MADRS scores were entered as the dependent variable, while Baseline pBDNF concentrations were entered as independent variables and other variables including age, gender, body weight, body mass index, psychiatric family history, previous hospitalization, psychiatric comorbidity, and age of onset were entered as covariate variables. Furthermore, an additional analysis was also performed on a subsample of patients with TRD in this study. IBM SPSS version 23 software (IBM Corporation, Armonk, NY, USA) was used in this study, and significance was set as p-value less than 0.05.

Results

Ninety-four individuals (aged 18 to 62 years) with unipolar or bipolar depression who provided a baseline blood sample were enrolled. Of these patients, 81.9% (77/94) fulfilled the diagnostic criteria of TRD. Baseline pBDNF concentrations with a mean value of 10.1 ng/ml, ranged from 0.9 to 27.2 ng/ml.

Treatment outcome and BDNF

After the last ketamine treatments, the rates of response and remission were 68.1% (64/94) and 51.1% (48/94), respectively. The rates of response and remission for patients with TRD were 68.8% (53/77) and 51.9% (40/77), respectively, after completion of six ketamine infusions. Higher baseline pBDNF concentrations were found in ketamine responders/remitters (11.0 ± 6.2/10.1 ± 5.8 ng/ml) than nonresponders/nonremitters (8.0 ± 5.5/9.2 ± 6.4 ng/ml) (all ps<0.05, Table 1).

Table 1 Comparison of baseline sample characteristics between responders and nonresponders and between remitters and nonremitters.

Variables	Total (n = 94)	Response after six ketamine infusions	Remission after six ketamine infusions	
		Responders (n = 64)	Nonresponders (n = 30)	Statistics	Remitters (n = 48)	Nonremitters (n = 46)	Statistics	
	N (%)	N (%)	N (%)	X2	p	N (%)	N (%)	X2	p	
Female	50 (53.2)	35 (54.7)	15 (50.0)	0.2	0.67	22 (45.8)	28 (60.9)	2.1	0.14	
Employment	38 (40.4)	29 (45.3)	9 (30.0)	2.0	0.16	22 (45.8)	16 (34.8)	1.2	0.28	
Married	50 (53.2)	35 (54.7)	15 (50.0)	0.2	0.67	27 (56.3)	23 (50.0)	0.4	0.54	
	Mean (SD)	Mean (SD)	Mean (SD)	T/Z	p	Mean (SD)	Mean (SD)	T/Z	p	
Age (years)	34.6 (11.6)	35.1 (11.2)	33.4 (12.5)	−0.7	0.50	34.8 (10.9)	34.3 (12.4)	−0.2	0.83	
Education (years)	12.4 (3.3)	12.8 (3.2)	11.4 (3.4)	−2.0	0.049	12.6 (3.2)	12.1 (3.4)	−0.6	0.53	
BMI (kg/m2)	22.4 (3.6)	22.5 (3.5)	22.2 (3.8)	−0.4	0.66	22.7 (3.9)	22.1 (3.2)	0.8	0.43	
Duration of illness (months)	102.5 (98.3)	106.3 (101.7)	94.5 (91.9)	—a	0.51	107.3 (101.4)	97.6 (95.9)	—a	0.58	
Baseline MADRS scores	31.9 (7.6)	31.8 (7.6)	32.2 (7.6)	0.3	0.80	30.6 (7.3)	33.3 (7.7)	−1.8	0.08	
pBDNF concentrations (ng/ml)	10.1 (6.2)	11.0 (6.2)	8.0 (5.5)	—a	0.01	10.1 (5.8)	9.2 (6.4)	—a	0.045	
Notes.

a Mann–Whitney U test.

Bolded values are p<0.05.

pBDNF plasma brain derived neurotrophic factor

BMI Body Mass Index

MADRS Montgomery-Asberg Depression Rating Scale

SD standard deviation

Linear mixed models showed that MADRS scores and pBDNF concentrations exhibited significant time main effects between responders and nonresponders and between remitters and nonremitters (Table 2). Ketamine produced a significant change in MADRS scores and pBDNF concentrations at 13 d and 26 d when compared to baseline (Figs. 1 and 2). Similar results were found in patients with TRD (Table S1, Figs. S1 and S2).

Table 2 Comparison of MADRS scores and pBDNF concentrations between responders and nonresponders and between remitters and nonremitters in patients with unipolar and bipolar depression using linear mixed model analysis.

Outcomes	Variables	Group-by-time interaction	Time main effect	Group main effect	
		F	p	F	p	F	p	
Responders vs. nonresponders	MADRS scores	59.79	<0.001	223.39	<0.001	59.32	<0.001	
pBDNF concentrations (ng/ml)	0.04	0.837	8.55	<0.001	3.90	0.024	
Remitters vs. nonremitters	MADRS scores	74.95	<0.001	263.13	<0.001	29.52	<0.001	
pBDNF concentrations (ng/ml)	0.02	0.888	6.40	0.003	2.61	0.079	
Notes.

Bolded values are p < 0.05.

pBDNF plasma brain derived neurotrophic factor

MADRS Montgomery-Asberg Depression Rating Scale

Correlation of BDNF and MADRS scores

Correlation analyses showed significant associations between pBDNF concentrations at baseline and MADRS scores at 13 d and 26 d in depressed patients (all p s<0.05; Table 3). The significant association of pBDNF concentrations at baseline and MADRS scores at 13 d (t = −2.011, p = 0.047) and 26 d (t = −2.398, p = 0.019) remained in multiple regression analysis. Similar results were found in patients with TRD (Table S2).

Figure 1 Change in depressive symptoms in patients with unipolar and bipolar depression.

# Significant difference was found when comparing baseline to the indicated times (p < 0.05). ∗ Significant difference was found between responders and nonresponders and between remitters and nonremitters at the indicated times (p < 0.05). Abbreviations: MADRS, the Montgomery-Asberg Depression Rating Scale.

Figure 2 Change in pBDNF concentrations in patients with unipolar and bipolar depression.

# Significant difference was found when comparing baseline to the indicated times (p < 0.05). ∗ Significant difference was found between responders and nonresponders and between remitters and nonremitters at the indicated times (p < 0.05). Abbreviations: pBDNF, plasma brain derived neurotrophic factor.

Discussion

This is the first study to determine pBDNF concentrations after six subanesthetic intravenous ketamine in Chinese individuals suffering from unipolar and bipolar depression and to investigate the correlation of pBDNF concentrations at baseline and six subanesthetic intravenous ketamine’s antidepressant efficacy. The following main findings included: (1) ketamine increased pBDNF at 13 d and 26 d compared to baseline; (2) responders/remitters had significantly higher baseline pBDNF concentrations than nonresponders/nonremitters; (3) MADRS scores showed significant improvement at both time points across the total sample compared to baseline; (4) baseline pBDNF concentrations were related with MADRS scores; and (5) additional analysis of patients with TRD also found that pBDNF concentrations were related with the antidepressant outcome of ketamine in patients with TRD.

Consistent with an animal study after single ketamine infusion (Pytka et al., 2018), our study demonstrated that ketamine increased pBDNF concentrations after six ketamine infusions. Although nonresponders/nonremitters had significantly lower pBDNF concentrations at baseline than responders/remitters, repeated ketamine infusions failed to significantly increase pBDNF concentrations in responders/remitters when compared to nonresponders/nonremitters. Similarly, a previous study found no changes in pBDNF concentrations in individuals suffering from TRD after completion of an intravenous infusion of ketamine compared to baseline (Machado-Vieira et al., 2009). However, Haile et al. found that pBDNF concentrations were significantly increased following a single ketamine infusion in responders compared to nonresponders (Haile et al., 2014). Therefore, these findings should be confirmed by randomized controlled trials.

Table 3 Correlation of baseline pBDNF concentrations and MADRS scores at 13 d or 26 d in patients with unipolar and bipolar depression.

Variables	MADRS scores at 13 d	MADRS scores at 26 d	
pBDNF concentrations (ng/ml)	r =  − 0.220	r =  − 0.278	
p=0.033	p=0.007	
Notes.

Bolded values are p<0.05.

pBDNF plasma brain derived neurotrophic factor

MADRS the Montgomery-Asberg Depression Rating Scale

r Pearson coefficient of correlation

The observed rapid reduction in MADRS scores lasted up to 2 weeks, replicating the previous findings (Rasmussen et al., 2013; Shiroma et al., 2014). However, the primary objective of this study is to examine the association of baseline pBDNF concentrations and six subanesthetic intravenous ketamine’s antidepressant efficacy. Several studies examined the association of pBDNF concentrations with the antidepressant response of a single infusion of ketamine, but these findings are inconsistent (Haile et al., 2014; Lee & Kim, 2010). For instance, one study reported that pBDNF concentrations were related with the severity of depression (Haile et al., 2014). However, Machado-Vieira et al.’s study reported a negative finding on the association of pBDNF concentrations and ketamine’s antidepressant efficacy (Machado-Vieira et al., 2009).

Notably, several animal studies reported that increased hippocampal and cortical BDNF expression can partly accounting for ketamine’s antidepressant-like efficacy (Autry et al., 2011; Réus et al., 2011). pBDNF concentrations were lower in individuals suffering from depression compared to healthy controls (Kishi et al., 2017; Munno et al., 2013) and increased after receiving antidepressants (Munno et al., 2013; Polyakova et al., 2015), electroconvulsive therapy (Luan et al., 2020; Piccinni et al., 2009), and repeated transcranial magnetic stimulation (Yukimasa et al., 2006). Therefore, neurotrophic factors, such as BDNF, might be involved in ketamine’s antidepressant mechanism. Notably, BDNF is implicated in the regulation of synaptic plasticity, including the synaptic recruitment of AMPA receptors. Growing studies indicate that synaptic plasticity is altered in individuals with depression (Machado-Vieira, Zarate Jr & Manji, 2006; Schloesser et al., 2008; Zarate Jr, Singh & Manji, 2006), and ketamine’s antidepressant efficacy may be attributed to the synaptic potentiation of neural circuits mediated by increased AMPA-to-NMDA glutamate receptors (Maeng & Zarate Jr, 2007).

The following limitations should be acknowledged. First, the participants continued receiving previous medications and lacked a washout period during the study, which may have affected pBDNF concentrations. However, the combination of ketamine and other antidepressants for individuals with depression is increasingly being used in the real-world clinical setting (Shiroma et al., 2014). Second, the sample size was small in the current study. Third, the possible impact of subjective evaluation was inevitable due to lack of a control group. Fourth, some comprehensive analyses, such as the mediating and moderating effect analysis, were not conducted in this study. Finally, brain BDNF concentrations and other key neurobiological mediators, such as mTOR, were not directly measured. However, BDNF crosses the blood–brain barrier, and pBDNF concentrations are closely correlated with cortical BDNF concentrations, and likely reflect brain BDNF concentrations (Pillai et al., 2010; Poduslo & Curran, 1996).

In conclusion, this preliminary study suggests that baseline pBDNF concentrations appeared to be correlated with ketamine’s antidepressant efficacy in Chinese patients with depression.

Supplemental Information

Supplemental Information 1 Raw data

Click here for additional data file.

Supplemental Information 2 Supplemental figures and tables

Click here for additional data file.

Supplemental Information 3 Trial protocol

Click here for additional data file.

Additional Information and Declarations

Competing Interests

Author Contributions

Human Ethics

Clinical Trial Ethics

Ethics

Data Availability

Clinical Trial Registration

The authors declare there are no competing interests.

Wei Zheng performed the experiments, prepared figures and/or tables, authored or reviewed drafts of the paper, and approved the final draft.

Yan-Ling Zhou conceived and designed the experiments, performed the experiments, authored or reviewed drafts of the paper, and approved the final draft.

Cheng-Yu Wang performed the experiments, authored or reviewed drafts of the paper, and approved the final draft.

Xiao-Feng Lan and Bin Zhang performed the experiments, authored or reviewed drafts of the paper, and approved the final draft.

Su-Miao Zhou and Su Yan analyzed the data, prepared figures and/or tables, authored or reviewed drafts of the paper, and approved the final draft.

Yu-Ping Ning conceived and designed the experiments, authored or reviewed drafts of the paper, and approved the final draft.

The following information was supplied relating to ethical approvals (i.e., approving body and any reference numbers):

The Affiliated Brain Hospital of Guangzhou Medical University granted ethical approval to carry out the study within its facilities (Ethical Application Ref: 2016-030).

The following information was supplied relating to ethical approvals (i.e., approving body and any reference numbers):

The Affiliated Brain Hospital of Guangzhou Medical University granted ethical approval to carry out the study within its facilities (Ethical Application Ref: 2016-030).

The following information was supplied relating to ethical approvals (i.e., approving body and any reference numbers):

The Affiliated Brain Hospital of Guangzhou Medical University granted ethical approval to carry out the study within its facilities (Ethical Application Ref: 2016-030).

The following information was supplied regarding data availability:

Raw data are available as a Supplemental File.

The following information was supplied regarding Clinical Trial registration:

ChicCTR-OOC-17012239

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
