# Peer review of "Plasma BDNF concentrations and the antidepressant effects of six ketamine infusions in unipolar and bipolar depression"

_PeerJ, doi:10.7717/peerj.10989_

## Round 0.1 · original submission · Major Revisions

The paper has a lot of issues to be addressed. The authors may consider to reconstruct the research question and do the corresponding analyses accordingly.

Reviewer 1 ·

Basic reporting

None

Experimental design

None

Validity of the findings

None

Additional comments

This paper conducted a study that have determined the association between plasma BDNF levels and the antidepressant effects of six ketamine infusions. Though this study could provide clinical implications. There are several problems to be addressed:
(1) As you mentioned in the introduction, the association of plasma BDNF levels and the antidepressant effects of repeated ketamine infusions in depressed patients is not clear. So why does not author explore the mediating/moderating effect in this manuscript? Maybe it's a clearer relationship.
(2) Although ninety-four depressed patients were included, sample size still is small. This is a limitation.
(3) How to define the bipolar depression? And how to distinguish the unipolar or bipolar depression? Authors should conduct a subgroup analysis to explore potential differences between unipolar depression and bipolar depression.
(4) For statistical analysis, authors did not adjust any covariates. Why? I think author should consider the potential confounds.
(5) Line 309. Authors should report the full name of mTOR.
(6) Authors should add some clinical implications in the discussion.
(7) Table 2 showed that both MADRS scores and plasma BDNF levels exhibited significant effects between responders and non-responders and remitters and non-remitter. Why does not author consider the interaction effect?

Reviewer 2 ·

Basic reporting

See my overall comments as below.

Experimental design

See my overall comments as below.

Validity of the findings

See my overall comments as below.

Additional comments

Some minor issues:

1. Line 60-61, response and remission can not be directly measured by MADRS.

2. Line 63-64, “baseline, 24h, and 2 weeks” do not correspond to “(baseline, 13d and 26d)”.

3. Line 69, please provide detailed data of the BDNF levels and line 71, the corresponding correlation coefficients.

4. Line 73, in the conclusion, the authors used “may be a predictor” but in the title BDNF predicts treatment response definitely.

5. Line 104-105, a problematic and unclear sentence.

6. Line 108, are should be is.

7. Line 134-136, an unclear sentence, “examined patients” looks like very strange.

8. Line 155, according to (3), all subjects are patients with TRD. Line 206, why there is a subsample of patients with TRD in the whole sample?

9. Line 191-194, these statistical analysis methods are not used for “analyze”. Please specify these are used for comparisons between groups.

In general, scientific papers must be written rigorously without obvious language and logic issues. The authors did poorly on this.

Major issues:

1. The introduction is poorly written. Line 115-120, it remains unclear why there is a need for efficacy predictor of ketamine treatment. Line 136-137, because no non-treatment group, pre- and post-comparison can not answer this research question: increase in BDNF due to ketamine administration. The second hypothesis has no supporting evidence, or is not suitable in theory because in the afore-reviewed studies, higher BDNF is associated with better treatment response. The major issue of this part is that the authors failed to define the research question, for example, BDNF as a predictor of treatment response, biomarker of treatment response, or mediator of treatment response? These are different and can not be discussed together. A significant logic error of the current hypothesis is, since baseline BDNF could predict the treatment outcome at the end of the observation and ketamine has been administered at baseline, it is not necessary to talk so much on ketamine.

2. Problematic statistical analytic strategies. For the predictive value of baseline BDNF on treatment outcomes, I suggest the authors to do adjustment analysis to ensure this predictive value is independent from other baseline characteristics, for example, in Table 1, responders and remitters all had lower baseline MADRS scores, so responders and remitters may be a subgroup of patients with less severe depressive symptoms and are more likely to have better treatment response. The authors need to consider the sample size, since this seems inadequate for the adjustment analysis. Line 200, it remains unclear what is stepwise regression models and the purpose of such analysis. Third, I do not agree with the linear mixed model, since response and remission should be outcome variables, not an independent variable in the regression analysis. If the authors focused on the mediating roles of BDNF, the correct statistical method should be cross-lagged effect model. The current analysis can not denote the predictive effect of BDNF, i.e., prior BDNF change predicts latter MADRS change.

---

## Round 0.2 · Minor Revisions

The paper looks much better than the previous one. Please kindly address the remaining issues. Also, please have the paper polished again before the resubmission.

Reviewer 1 ·

Basic reporting

None

Experimental design

None

Validity of the findings

none

Additional comments

Thank you for your revision. I still have some comments for your consideration.

1. Title. It is well-known that BDNF can be considered as both BDNF gene and plasma BDNF level. Please specifically define it.
2. Please add the provider of SPSS software
3. Line 196-120. This is an inaccurate description. The main function of multiple linear regression is to explore the independent association, rather than controlling the covariate. Authors should re-structure this description and add the dependent variable and independent variables.

Reviewer 2 ·

Basic reporting

I have no further comments.

Experimental design

I have no further comments.

Validity of the findings

I have no further comments.

Additional comments

I have no further comments.

---

## Round 0.3 · accepted · Accept

Thanks for the revisions.

Reviewer 1 ·

Basic reporting

None

Experimental design

none

Validity of the findings

None

Additional comments

I am satisfied with your revision